# Through the Looking Glass of Social Media. Focus on Self-Presentation and Association with Mental Health and Quality of Life. A Cross-Sectional Survey-Based Study

**DOI:** 10.3390/ijerph18063319

**Published:** 2021-03-23

**Authors:** Jens Christoffer Skogen, Gunnhild Johnsen Hjetland, Tormod Bøe, Randi Træland Hella, Ann Kristin Knudsen

**Affiliations:** 1Department of Health Promotion, Norwegian Institute of Public Health, 5015 Bergen, Norway; gunnhildjohnsen.hjetland@fhi.no (G.J.H.); ann.kristin.knudsen@fhi.no (A.K.K.); 2Centre for Evaluation of Public Health Measures, Norwegian Institute of Public Health, 0213 Oslo, Norway; 3Alcohol and Drug Research Western Norway, Stavanger University Hospital, 4010 Stavanger, Norway; 4Faculty of Health Sciences, University of Stavanger, 4021 Stavanger, Norway; 5Department of Psychosocial Science, Faculty of Psychology, University of Bergen, 5020 Bergen, Norway; tormod.boe@uib.no; 6Department of Work, Social Services and Housing, Bergen Municipality, 5020 Bergen, Norway; randi.hella@bergen.kommune.no; 7Centre for Disease Burden, Norwegian Institute of Public Health, 5015 Bergen, Norway

**Keywords:** social media use, self-presentation, mental health, well-being, anxiety, depression, adolescence

## Abstract

Social media (SOME) use among adolescents has been linked to mental health and well-being. SOME self-presentation has been highlighted as an important factor to better understand the potential links. The aims of this study were to investigate the association between focus on SOME self-presentation and mental health and quality of life among adolescents. We used a cross-sectional survey, with *n* = 513 (56%; mean age 17.1 years; 58% boys) students from a senior high school in Norway. Associations between focus on SOME self-presentation and symptoms of anxiety and depression and quality of life were investigated using blobbograms, standardized mean difference (SMD), and gender-specific linear regression models. A high focus on SOME self-presentation was associated with more mental health problems and reduced quality of life. The strength of the associations with symptoms of depression (0.75SMD) and anxiety (0.71SMD) was large, while it was medium-large for quality of life (−0.58SMD). The association was similar across gender in relation to symptoms of anxiety. For symptoms of depression and quality of life, the association was stronger for girls compared to boys. Our findings yield preliminary evidence of a potential relationship between focus on SOME self-presentation and mental health.

## 1. Introduction

Adolescence is an important period in life characterized by increased independence from parents and an expanded social life dominated by one’s peers [1]. It is also a period with increased emotional upheaval [2], and many mental health problems commonly emerge during adolescence [1]. There is also evidence that suggest quality of life is lower during adolescence compared to earlier childhood years [3].

Today’s adolescents grow up in the age of social media use and online communication. Social media can be defined as “highly interactive platforms via which individuals and communities share, co-create, discuss, and modify user-generated content” [4]. Social media use among adolescents has been linked to mental health and well-being [5], and some studies indicate that social media use is associated with more mental health problems and decreased well-being. For example, studies have found that more time spent on social media is associated with symptoms of depression and anxiety [6,7,8], conduct problems, and episodic heavy drinking [6]. However, by focusing primarily on frequency and duration of use [5,9,10], these studies provide little knowledge of how various types of social media activities may be differentially linked to mental health and well-being. Although a growing number of studies in general indicate that social media use should be considered multifaceted, and that distinct uses is differentially associated with adverse outcomes (see for instance [11,12,13,14]), there is still a dearth of studies taking this into consideration. Thus, there is a call for more novel quantitative studies investigating the relationship between what adolescents do on social media and how this is associated with mental health and well-being.

Self-presentation has been highlighted as one important social media activity by many researchers [15,16,17,18,19]. Self-presentation may be defined as individual practices related to how one present oneself to others [20] and is generally thought to be motivated by a wish to make a socially desirable impression on others, while still remaining true to one’s own beliefs and ideals.

Self-presentation practices on social media include sharing of self-created content, posting of opinions, and promoting online content that one is interested in (like news, music, and movies), and adolescents are reported to be more engaged in these activities than any other age-group (see instance Herring and Kapidzic [18]). Social media gives adolescents control over what, where (on which platform), how, and when they self-present to others. The immediacy of social feedback (e.g., likes and comments) provides cues about social desirability and direction for adjusting future presentations to be in alignment with how adolescents ideally would like to present themselves. These feedback mechanisms and other features of social media, such as the potential to reach multiple audiences, seems to facilitate self-presentation on social media [21]. Given the focus many social media platforms puts on likes, other "nano-level interactions" [22], number of followers, and comments, it seems likely that issues related to how one present oneself online is a potent factor in many adolescents’ life. This may be further accentuated by engagement in social comparison on social media [23].

A few previous studies have shown that activities of self-presentation on social media are associated with well-being and mental health. Frison and Eggermont found, for instance, that adolescents that actively self-presented by posting content on social media reported fewer depressive symptoms compared to those who used social media to more passively observe (i.e., consume) other’s content [24]. Similarly, a recent longitudinal study following participants at 10, 12, and 14-years-of age, reported that more passive, “other-oriented” social media use (i.e., merely commenting or “liking” others’ posts) versus actively self-presenting on social media had a differential effect on self-esteem related to physical appearance [25]. Specifically, the authors found that increased other-oriented social media use was associated with reduced subsequent appearance self-esteem, but this was not found for actively engaging in self-presenting on social media. They did, however, also report important gender differences. In gender-specific analyses they found that the relationship between earlier other-oriented social media use on later appearance self-esteem was strong among girls but absent among boys [25]. They also reported that girls were consistently more likely to engage in other-oriented social media use across all time points. Several other authors have also reported that there are gender differences in relation to self-presentation on social media [18,26,27]. Herring and Kapidzic reported for instance that adolescent girls are more likely to present their friendship ties on social media, while boys post updates more related to technology, sports, and humor [18]. Adolescent boys are also more likely to join or identify with online social groups that differ from their offline social groups [18]. Furthermore, adolescent girls use social media more to communicate with their peers and maintain or reinforce preexisting relationships, while boys on the other hand more actively seek out new people and make new friends online [18].

Previous studies also indicate a relationship between the authenticity of the self-presentation and mental health and well-being [9,28]. In 2017, Twomey and O’Reilly authored a systematic review of self-presentation on Facebook and mental health and personality [28]. They identified 21 studies of mostly adult and young adult participants, and among other things, found support for associations between false self-presentation and low self-esteem and higher levels of social anxiety, and true self-presentation and increased levels of self-esteem. In relation to primary studies, Jang and colleagues reported that a true self-presentation style led to greater self-reported happiness among high self-esteem users using an experimental approach in an adult sample [29]. Conversely, the use of instrumental or strategic (i.e., only presenting your “best self”) self-presentation were associated with more self-reported happiness regardless [30] of self-esteem level [29]. Additionally, among college students, strategic self-presentation has been reported to be associated with increased subjective well-being [31]. Furthermore, Reinecke and colleagues found that authenticity in self-presentation on social media was longitudinally associated with increased positive affect and decreased negative affect among adults [32]. In one of the few studies of adolescent participants, Xie and colleagues reported that online authentic self-presentation was associated with lower levels of depression symptoms as part of a mediation model [33]. On the other hand, false self-presentation behaviors is reported to be associated with negative mental health among adults, as indicated by higher symptom levels of depression, anxiety, and stress [34].

Although there are some studies investigating the potential consequences of different aspects of self-presentation on social media, most previous studies have focused on the antecedents or motivations for self-presentation [29]. Most of the studies we were able to identify, however, with the exception of Xie and colleagues [33], consisted of young adult or adult samples. This is also reflected by the included primary studies in Twomey and O’Reilly’s systematic review [28]. Furthermore, a recent narrative review highlighted that self-presentation may be an important factor to investigate in order to better understand the link between use of social media and well-being among adolescents [9]. Based on these considerations, there is a need for further studies investigating aspects of self-presentation on social media and associations with mental health and well-being among adolescents. Based on previous studies indicating the relevance of personal investment and emotional involvement in relation to social media use and mental health [35], one important aspect could be the focus (i.e., focus in terms of the attention and importance the adolescents give to their self-presentation activities) adolescents put on self-presentation on social media. We are not aware of any previous studies that have investigated the association between focus on self-presentation on social media and mental health and well-being.

The aims of the present study were to investigate the association between focus on self-presentation on social media and mental health and quality of life among adolescents. Specifically, we aimed to investigate the overall and gender-stratified associations between self-presentation and symptoms of depression and anxiety, as well as a general measure of quality of life.

## 2. Materials and Methods

### 2.1. Study Design, Setting, and Participants

The present study employs cross-sectional data from an online survey of adolescents recruited from a senior high school in Vestland County, Norway. All students aged 16 or more enrolled at the high school were invited to the survey, and *n* = 513 (56%) participated. The data were collected in March 2020. The students were first informed about the study from their teacher one day prior to participation, and the next day they received a survey-specific web address (uniform resource locator; URL) containing written online information about the study as well as the possibility to consent to participate. One school hour was allotted to completing the questionnaire. Four participants did not complete the survey and were therefore excluded from the present study. The mean age of the participants was 17.1 years (1.1 standard deviations), and 58% were boys. The study was approved by the regional ethics committee and is in agreement with the General Data Protection Regulation (See also “Institutional Review Board Statement” below for more information).

### 2.2. Variables

Age and Gender Were Reported by the Participants

#### 2.2.1. Independent Variables: Focus on Self-Presentation on Social Media

The majority of the questions regarding focus on self-presentation on social media were generated using an inductive approach based on qualitative interviews [36], while one question came from an existing survey [37]. The remaining items developed were based on four focus group interviews with adolescents related to use of social media and mental health. The transcripts of the interviews were analysed using thematic analysis, and items were generated to cover the identified themes (more details can be found in the online preprint: https://www.researchsquare.com/article/rs-133641/v1 (accessed on 19 March 2021)). To create items that sounded familiar to the respondents (of the questionnaire), items were worded as similar to what the interview subjects expressed as possible. The use of focus groups ensure that the instrument is relevant to the population in question and that issues considered important for adolescents were not left out, i.e., they ensure the content validity of the instrument [38]. Candidate items were reviewed and discussed by members of the research team before being included in the questionnaire. Next, a small group of psychology students and a resource group of consisting of adolescents completed a draft of the questionnaire and provided oral feedback to the proposed questions. The questionnaire was revised according to their feedback, which mainly pertained to the wording of some of the items. One of the emergent sub-themes identified was self-presentation on social media. Participant’s focus on self-presentation on social media was measured by seven questions:I use a lot of time and energy on content I post on social media.It is important for me that my posts receive many likes and/or comments.It is important for me to have many followers on social media.I delete posts on social media that do not receive enough likes and/or comments.I retouch pictures of myself to look better before I post them on social media.What other’s post (images/status updates/stories) makes me less satisfied with myself and my life.The response I get for what I post (images/status updates/stories) impacts how I feel.

The response categories were “not at all”, “very little”, “sometimes/partly true”, “a lot” and “very much”, coded 1 to 5. The questions are used separately and as a summed mean score (Cronbach’s alpha of 0.88) in the present study.

The participants were also asked which social media platforms they used, where they could choose from a list of 19 platforms: “Facebook”, “Instagram”, “Snapchat”, “LinkedIn”, “Twitter”, “YouTube”, “Pinterest”, “Messenger”, “WhatsApp”, “Jodel”, “TikTok”, “Discord”, “Twitch”, “Spond”, “Houseparty”, “Reddit”, anonymous platforms (for instance “Ask”, “Sarahah”), dating platforms (for instance “Tinder”, “Match”, “Grindr”, “Yubo”), and “other”. The participants could indicate as many of these platforms as necessary.

#### 2.2.2. Dependent Variables: Measures of Mental Health and Quality of Life

##### Symptoms of Anxiety

Anxiety was measured by the questionnaire General Anxiety Disorder 7 (GAD-7; Spitzer, Kroenke [39]). GAD-7 consists of 7 questions regarding symptoms of general anxiety scored from 1 (not at all) to 4 (almost every day). The questionnaire can be used as a continuous measure (total score, ranging from 0 to 21 in this study (28 maximum)) or as a dichotomous variable with a cut-off of 10. Cronbach’s alpha was 0.89 in the present sample.

##### Symptoms of Depression

Depression was measured by the questionnaire Short Mood and Feelings Questionnaire (SMFQ; Turner, Joinson [40]). SMFQ consists of 13 statements related to symptoms of depression with the following response options 0 (not true), 1 (sometimes true), and 2 (correct). The questionnaire can be used as a continuous measure (total score, ranging from 0 to 26) or as a dichotomous variable with a cut-off at the 90th percentile. Cronbach’s alpha was 0.91 in the present sample.

##### Quality of Life

Quality of Life (QoL) was measured using the Warwick–Edinburgh Well-being Scale (WEMWBS; Tennant, Hiller [41]). WEMWBS consists of 14 statements related to well-being and quality of life scored from 1 (not at all) to 5 (all the time). WEMWBS can be used as a continuous measure (total score, ranging from 5 to 70), and as a dichotomous measure with median split as the cut-off point. Cronbach’s alpha was 0.92 in the present sample.

### 2.3. Statistical Analyses

Descriptive gender-specific statistics of the independent and dependent variables are presented in a table. For the first figure, the associations between each self-presentation item and symptoms of anxiety and depression and quality of life are presented as blobbograms [42]. Blobbograms were used as they provide information about individually estimated associations, the precision of the estimates, and statistical significance, as well as an overall effect estimate in a succinct manner. This approach is widely used in meta-analysis but also in health research studies in general [42]. For the first figure the mean of each self-presentation item for those below and above the cut-offs for anxiety, depression and quality of life was calculated, as well as the standardized mean difference.

In order to assess the structural validity of the items related to focus on self-presentation, exploratory (EFA), and confirmatory factor (CFA) analysis was used to examine the underlying structure. Maximum likelihood (ML) estimation was used for CFA, and model fit was assessed by the comparative fit index (CFI), the root mean square error of approximation (RMSEA), and the standardized root mean square residual (SRMR). A CFI of 0.95 or greater, RMSEA of 0.06 or less, and SRMR of 0.08 or less were collectively considered an indication of good model fit to data [43].

The overall standardized mean difference for self-representation as a summed mean score is presented for each dependent variable. In the second figure, the predicted mean gender-standardized score for symptoms of depression and anxiety and quality of life (QoL) is presented across integers of the summed mean score of self-presentation for boys and girls separately. Linear gender-specific regressions were calculated to obtain the coefficient for the association between the mean summed score of self-presentation and the dependent variables. The mean gender-standardized score for symptoms of anxiety and depression, and QoL was obtained by a gender-stratified Z-scoring of the summed self-presentation variables (total score). Thus, each gender-specific bar represents deviance in standard deviations from overall gender-specific mean across focus on self-presentation, and the error bars denotes 95% confidence intervals. In the third figure, the ten most frequently used social media platforms overall are presented across quartiles of the summed score of self-presentation. Differential proportions of used platforms across quartiles of self-presentation were analysed using logistic regression models while adjusting for gender, and in gender-stratified analyses. The gender-adjusted and gender-specific trend across the rank of quartiles are presented as average odds ratios.

The variables included in the present study had some missing values, ranging from *n* = 3 to *n* = 48. To retain the maximum level of information in the analyses employed in Figures 1–3, we employed pairwise deletion to handle missingness. The statistical analyses were performed using Stata/SE 15.0 for Windows [44], R version 3.6.0 [45] and RStudio version 1.2.1335 [46] including the R-packages “gtsummary” version 1.3.6 [47], “lavaan” version 0.6.8 [48], and “meta” version 4.12.0 [49] including dependencies.

## 3. Results

### 3.1. Separate Items and Overall Standardized Mean Difference

Overall, girls indicated more focus on self-presentation on social media except for retouching of pictures of themselves (*p* = 0.448 for gender difference, see Table 1). For boys, the proportion for at least some focus on self-presentation ranged from 4% (“response important”) to 20% (“time and energy”), and the corresponding range for girls were 13% (“retouching”) and 46% (“time and energy”). Girls were also more likely to be above the cut-off for anxiety and depression, while being below the cut-off for QoL. In relation to the association between self-presentation and anxiety, case-level anxiety was associated with higher mean levels across each self-presentation item (Figure 1). The overall standardized mean difference was 0.71 (CI95% 0.47, 0.94). For depression, case-level depression was associated with higher mean levels for all but one item (retouching of pictures) of self-presentation. The overall standardized mean difference was 0.75 (CI95% 0.44, 1.06). For quality of life, high QoL (above median) was associated with lower mean levels for all items of self-presentation (overall standardized mean difference −0.58 (CI95% −0.76, −0.40)).

### 3.2. Structural Validity of Self-Presentation Items

Exploratory factor analysis of the seven items related to focus on self-presentation on social media indicated a unidimensional scale, with an eigenvalue of 3.7 for factor 1 and a drop to 0.4 for factor 2. The Kaiser–Meyer–Olkin measure of sampling adequacy was 0.86, and factor loadings ranged from 0.49 to 0.90, with a mean of 0.72. CFA was performed for a unidimensional scale, and adequate fit was obtained for a one-factor-solution (CFI: 0.990, RMSEA: 0.055 (90% Confidence intervals 0.029–0.080), SRMR: 0.029) with correlated error terms between item 6 and 7 (correlation 0.40). Based on these assessment the seven items were used as a summed score to represent overall focus on self-presentation on social media.

### 3.3. Overall Focus on Self-Presentation and Symptoms of Anxiety and Depression and Quality of Life

For both girls and boys there was a significant positive association between focus on self-presentation on SOME and symptoms of anxiety and depression (Figure 2). Having a high focus on self-presentation on SOME (“very much”) was associated with the highest predicted anxiety-symptom levels (1.2 SD and 1.1 SD for boys and girls, respectively) compared to the gender-specific mean. The average regression coefficient (trend) were comparable for boys (B: 0.35 SD, *p* < 0.001) and girls (B: 0.38 SD, *p* < 0.001).

The same patterns were found for depression symptom-levels (1.0 SD and 1.5 SD for boys and girls, respectively). The average regression coefficient was 0.30 SD (*p* < 0.001) for boys, and somewhat larger for girls (B: 0.53 SD, *p* < 0.001).

For quality of life, increased focus on self-presentation on social media was associated with decreased levels for girls (−1.2 SD for those with a high focus) and boys (−0.7 SD for those with high focus). The average regression coefficient was −0.21 SD (*p* = 0.012) for boys compared to −0.42 SD (*p* < 0.001) for girls.

### 3.4. Use of Different Platforms across Overall Focus on Self-Presentation

Figure 3 present the proportions of social media platforms used by the participants across quartiles of focus on self-presentation. Overall, there were modest differences across the quartiles. Statistically significant positive trends across the rank of quartiles of self-presentation were found for use of Instagram, Facebook, and TikTok, while a significant negative trend was found for Discord. In gender-specific analyses positive trends were found for Instagram and TikTok and negative trends were found for Discord for both girls and boys. For boys, the use of Instagram increased from 82.0% in the lowest quartile group to 95.1% in the highest quartile group, while use of TikTok went from 50.4% to 73.2% and use of Discord decreased from 52.3% to 31.7%. For girls, the corresponding differences were 84.6% to 100% for use of Instagram, 38.5% to 82.9% for use of TikTok, and 26.9% to 6.6% for use of Discord.

## 4. Discussion

The present study investigated the association between focus on self-presentation on social media and mental health and quality of life among adolescents. Overall, a high focus on self-presentation was associated with more mental health problems and reduced quality of life. The strength of the associations with symptoms of depression and anxiety was large, while it was medium to large for quality of life according to suggested guidelines on magnitude of effect [50]. In the gender-stratified analyses, the association was similar for boys and girls in relation to symptoms of anxiety. For symptoms of depression and quality of life, the association was stronger for girls compared to boys. In relation to the overall proportion of different platforms used across quartiles of focus on self-presentation, the differences were mostly modest. We did, however, find an increased proportion of use of Instagram, Facebook, and TikTok with increasing focus on self-presentation, and a corresponding decreased use of Discord. For both girls and boys, increased focus on self-presentation was associated with increased proportion of use of the highly visually focused platforms Instagram and TikTok, and decreased use in the gaming-related platform Discord.

In sum, our findings are preliminary evidence of a robust relationship between focus on self-presentation on social media and symptoms of anxiety and depression for both genders, albeit with potential important gender differences. This is in accord with recently reported associations between types of social media use and mental health indicators among adolescents and young adults [25,51]. Although we report an association between focus on self-presentation on social media and poor mental health and lower quality of life, we cannot assert the direction of the association. So, while it may be that increased focus on self-presentation is related to poorer mental health, it may also be true that pre-existing mental health problems lead to an increased focus on self-presentation. It is also just as likely that both relationships co-exist, so that both factors form part of a self-reinforcing cycle, which lead to both increased focus on self-presentation and poorer mental health.

Importantly, the present study investigated to what degree self-reported attention to self-presentation on social media and the importance of feedback was associated with mental health, rather than just how much time the participants spent engaged in self-presentation behavior. Our results do not necessarily indicate that engaging in self-presentation behavior per se is associated with poor mental health and well-being, but rather that the focus on these aspects of social media are important factors to gauge. Although the frequency and duration of self-presentation behaviors is likely to be closely related to the time spent engaging in these behaviors, it is also possible that adolescents may engage in self-presentation behavior without caring much about the feedback they receive. In fact, Frison and Eggermont [24] found that adolescents that actively presented themselves on social media reported fewer depressive symptoms than those who did not. Thus, self-presentation on social media in itself may not be harmful, but preoccupation with presenting oneself in a manner that elicit a wanted feedback from others may be detrimental to mental health. Differential associations with mental health and well-being is also likely to be related to true versus strategic self-presentation practices as reported by Jang and colleagues [29]. This notion is further echoed by a qualitative study of adolescent girls which reported on the importance put on self-presentation on social media in relation to self-esteem and insecurity by the participants [52]. The participants reported, for instance, that they would delete posted photographs with few likes out of frustration or embarrassment.

Only one of the items used in the present study explicitly asked about appearance-related self-presentation (retouching selfies to look better), while the other items asked about content and reactions in general. One underlying mechanism for the negative association between self-presentation and mental health found in this study may be related to appearance-related self-presentation. In line with this, a recent study found that social media engagement and behaviors involving appearance comparisons and judgements thereof is of particular importance for symptoms of depression and anxiety among adolescents and young adults [51]. This may be due to the immediacy of comparison facilitated by pictures as compared to other media [53], as well as the importance put on—females’ in particular—appearance and attractiveness emphasized in most cultures [54]. Furthermore, posting pictures of themselves may leave them more vulnerable to negative feedback or lack of feedback. In line with this, we found that increased focus on self-presentation was associated with an increased use of the highly visually-focused platforms Instagram and TikTok.

Self-presentation on social media was recently highlighted as a potentially important part of the puzzle to increase our understanding of the relationship between social media use and mental health and well-being among adolescents [9]. In our sample, a relatively large proportion reported at least some focus on self-presentation on social media, but for six out seven indicators (except “retouching”), the girls reported more focus on these aspects compared to boys. This is in line with other findings, where adolescent and young adult women report higher preoccupation with self-presentation on social media compared to boys [51], but are somewhat at odds with findings from younger age groups, where boys and girls displayed similar levels of self-oriented social media activity [25]. A study investigating selfie-taking and posting patterns found, however, consistent gender differences across broad age groups (aged 12 to 50 years). In that study, adolescents (aged 12 to 19 years) were found to be more likely than older aged groups to take own- and group-selfies, post their own selfies, and use filters. Furthermore, females were in general more likely to take personal and group selfies, post personal selfies, crop photos, and use filters compared to males, a gender-difference that was more pronounced during in the adolescent group [55]. Based on these findings, it may be that one of the underlying mechanisms for our observed gender differences the relationship between self-presentation and mental health are due to differences in the selfie-culture between boys and girls. If so, the differences in selfie-culture must be understood in light of the prevailing general gender culture, where female physical attractiveness is highly valued [56].

The present results should also be juxtaposed to findings suggesting a link between false self-presentation and poor mental health [28]. It is possible that adolescents who are preoccupied with self-presentation and the feedback they receive on social media are more likely to present themselves in a way that generates positive feedback, which may not correspond to their true self. This may be particularly true for use of social media platforms that very much are based on visual self-presentation, and thus may confer more use of retouching of self-portraits.

### 4.1. Implications and Public Health Relevance

For adolescents, our findings highlight the importance of awareness regarding how different types of activities on social media may be related to mental health outcomes. Today’s adolescents are social media "natives" [57], i.e., they have grown up in an “online world”. In contrast, many adults are “social media immigrants”, and this native/immigrant distinction may reduce parents’ understanding of and insight into adolescents’ online lives [58]. Parental involvement in relation to social media use has been reported to be stronger in younger ages [59]. Continued parental involvement during late childhood and early adolescence may therefore be important to mitigate potential negative effects of social media use [59,60]. Gómez and colleagues encourages for instance the empowerment of parents to moderate their offspring [59]. Similarly, the schools and teachers may play an important role in increasing awareness and knowledge about the use of social media. Chua and Chang suggest that educational programs that increase social media literacy can be beneficial when they also target negative consequences due to excessive peer comparison and the need for online feedback [52].

Our findings suggest that not only the amount of use on social media should be considered but also the role of self-presentation. A recent review covering the association between social media use and well-being among adolescents highlighted among other things the negative association between well-being and social media use through high investment and negative feedback [35]. They also asserted the need for intervention programs and educational programs that could address potential risks of social media relation to subjective well-being [35]. Specifically, they listed improved education and awareness among adolescents themselves, as well as the school as an optimal place to facilitate critical thinking about social media use and potential consequences.

### 4.2. Strengths and Limitations

The present study holds some strengths. First, the measures of mental health and quality of life are widely used and validated scales [39,40,41]. Second, the questions related to focus on self-presentation on social media were specifically developed for the purposes of the survey. Specifically, the questions related to focus on self-presentation were derived from four focus-groups where social media, mental health, and well-being was the overarching topic. Third, the survey is recent and included both girls and boys. Several limitations are also important to mention. First, the study is cross-sectional, and causality cannot be inferred. Second, the sample size is relatively small, limiting the meaningfulness of subgroup analyses, and the inclusion of potential confounders and moderators in our analyses due to lack of statistical power. For instance, the finding of a non-significant association between retouching and symptoms of depression may reflect the true relationship between the variables, be due to limited statistical precision, or due to confounding factors. Third, the survey did not include measures of frequency or duration of social media use among adolescents. However, such measure are likely to be inaccurate and biased [61,62], and may be non-germane [5,9,10,25]. It should also be noted that we were not able to differentiate between the frequency and duration of use of different social media platforms, which would have been beneficial in the comparison of platforms used across levels of self-presentation. For instance, over 97% of the sample reported any use of Snapchat, but how much they use this, and other platforms, probably varies substantially between different subgroups. Fourth, the study population was limited to one senior high school in Norway, which is likely to reduce the generalizability of our findings. Related to this, the participation rate was moderate, and this may impact the validity of our findings vis-à-vis the study population. However, several studies have indicated that low participation rate is more detrimental to prevalence estimates than estimates related to associations between variables [63]. Fifth, only a small proportion of the sample reported retouching pictures, and given the general availability of different photo-filters in different social media apps, our numbers seem low. Although we cannot confirm it, there is a strong possibility that this question was taken as editing of photos which goes beyond built-in filters and more along the lines of manually editing particular parts of the photos.

### 4.3. Future Research and Scope

Given the preliminary nature of the present findings, our findings needs to be investigated in other samples. Specifically, there is a need to investigate the proposed associations in larger and more diverse study samples. This would increase the precision of the estimates as well as enable investigations into potential confounding and interaction effects. Longitudinal data would also be beneficial, as this could get us a step closer to disentangling the temporal association between self-presentation and mental health and well-being, as well as open up for an exploration of mediation effects. The present study demonstrate the advantage of considering specific aspects of social media use, unravelling the diverse effects of social media use as indicated by conflicting previously reported study results [30].

## 5. Conclusions

The present study is the first study we are aware of that have examined the association between focus on self-presentation on social media and mental health and quality of life among adolescents. The study contributes with new knowledge about a specific aspect of social media use, highlighted as potentially particularly important to increase our understanding of the link between social media use and mental health among adolescents. Our findings indicate that there is a medium-strong and consistent relationship between focus on self-presentation and symptoms of anxiety and depression and quality of life among adolescents, and that there may be gender differences. The reported relationship could have important public health implications for adolescents, especially due to the way many social media platforms actively encourages a focus on self-presentation, social comparison, and presentations of the “perfect life”. Due to the limitations of the present study, future studies should investigate the relationships reported in larger study samples, in more diverse samples of adolescents and in a longitudinal rather than cross-sectional design.

## Figures and Tables

**Figure 1 ijerph-18-03319-f001:**
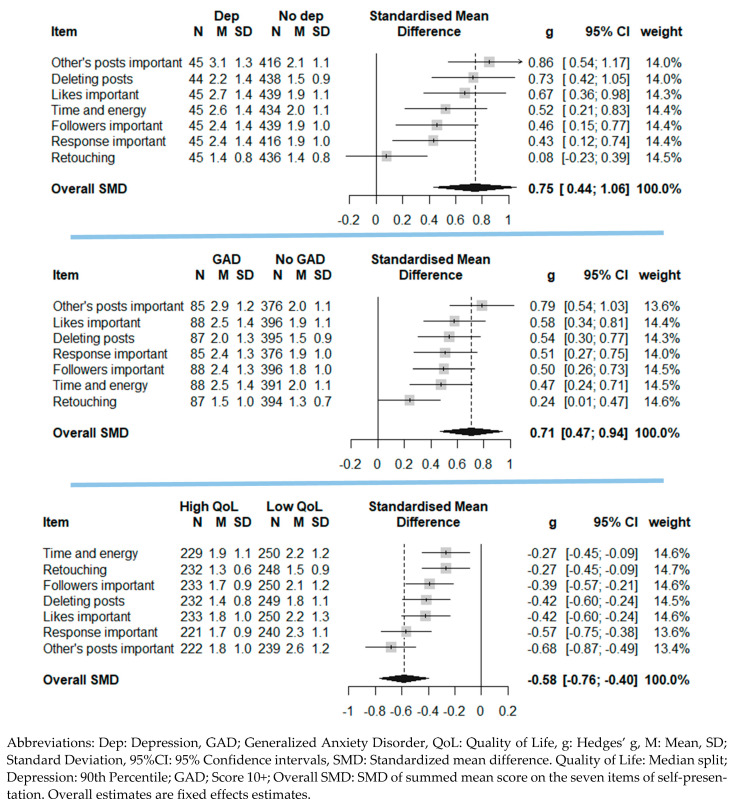
Associations between quality of life, depression and generalized anxiety, and self-reported focus on self-presentation on social media presented as blobbograms. Horizontal lines represent 95% confidence intervals, while the black diamond represents the overall standardized difference.

**Figure 2 ijerph-18-03319-f002:**
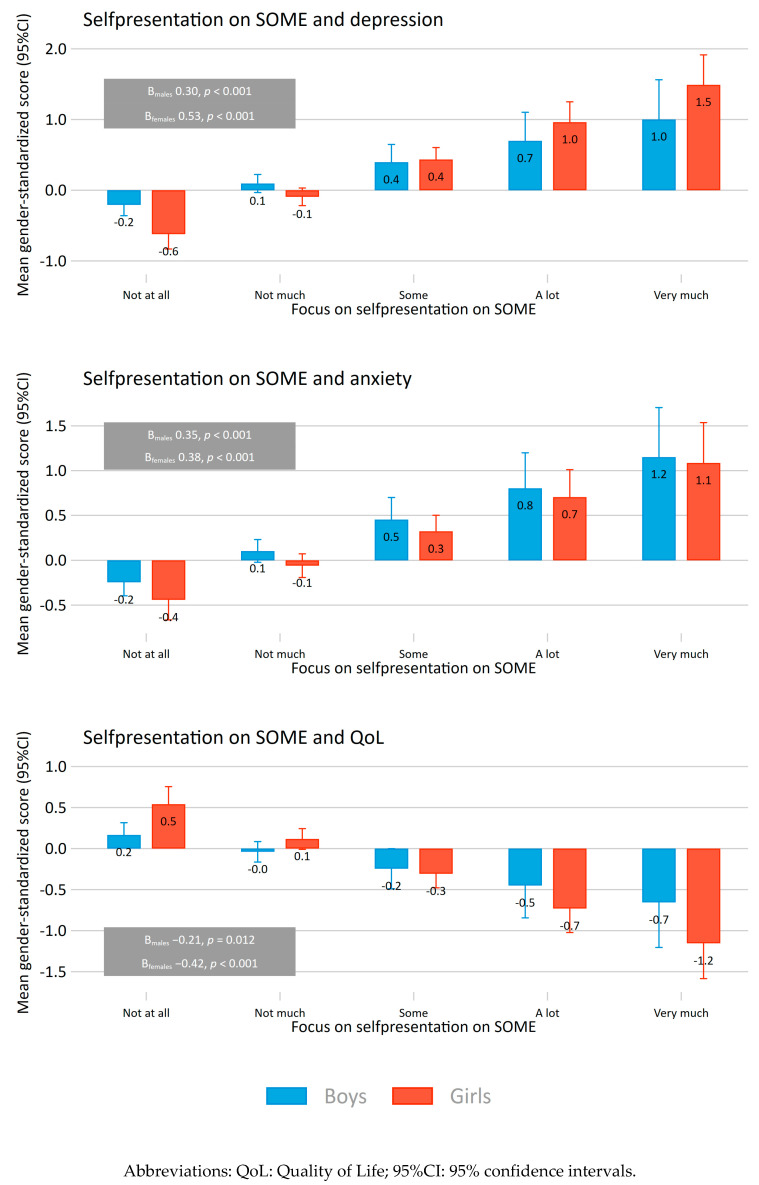
Associations between quality of life, depression and generalized anxiety, and self-reported focus on self-presentation on social media. Gender-stratified. Bars denote predicted deviation from gender-specific mean for symptoms of depression and anxiety and quality of life with 95% confidence intervals. Standardized score for average gender-specific coefficients (trend) in textboxes.

**Figure 3 ijerph-18-03319-f003:**
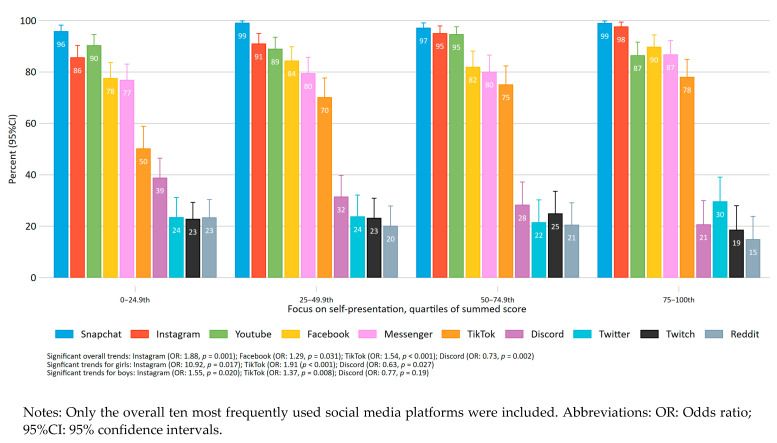
Gender-adjusted proportions of social media platforms used by the participants across quartiles of focus on self-presentation.

**Table 1 ijerph-18-03319-t001:** Description of included study variables across gender. *n* = 509.

Variables	Boys, *n* = 296 ^1^	Girls, *n* = 213 ^1^	*p*-Value ^2^
Time and energy			<0.001
Not at all	146 (53%)	57 (28%)	
Not much	71 (26%)	51 (25%)	
Some	37 (13%)	55 (27%)	
A lot	16 (5.8%)	29 (14%)	
Very much	6 (2.2%)	11 (5.4%)	
Missing	20	10	
Likes important			<0.001
Not at all	161 (57%)	63 (31%)	
Not much	61 (22%)	55 (27%)	
Some	40 (14%)	42 (21%)	
A lot	12 (4.3%)	32 (16%)	
Very much	6 (2.1%)	12 (5.9%)	
Missing	16	9	
Followers important			<0.001
Not at all	162 (58%)	62 (30%)	
Not much	69 (25%)	65 (32%)	
Some	35 (12%)	40 (20%)	
A lot	9 (3.2%)	29 (14%)	
Very much	5 (1.8%)	8 (3.9%)	
Missing	16	9	
Deleting posts			0.005
Not at all	209 (75%)	119 (59%)	
Not much	39 (14%)	40 (20%)	
Some	20 (7.1%)	23 (11%)	
A lot/Very much ^a^	12 (4.3%)	20 (9.9%)	
Missing	16	11	
Retouching			0.4
Not at all	220 (79%)	149 (74%)	
Not much	37 (13%)	27 (13%)	
Some	15 (5.4%)	18 (8.9%)	
A lot/Very much ^a^	7 (2.6%)	8 (4.0%)	
Missing	17	11	
Other’s post important			<0.001
Not at all	140 (53%)	37 (19%)	
Not much	58 (22%)	34 (17%)	
Some	57 (21%)	79 (41%)	
A lot	6 (2.3%)	26 (13%)	
Very much	5 (1.9%)	19 (9.7%)	
Missing	30	18	
Response important			<0.001
Not at all	147 (56%)	55 (28%)	
Not much	59 (22%)	59 (30%)	
Some	48 (18%)	52 (26%)	
A lot	5 (1.9%)	23 (12%)	
Very much	5 (1.9%)	8 (4.1%)	
Missing	32	16	
SMFQ, 90th percentile			<0.001
Below	279 (94%)	181 (85%)	
Above	17 (5.7%)	32 (15%)	
GAD-7, score 10+			<0.001
Below	258 (87%)	156 (73%)	
Above	38 (13%)	57 (27%)	
WEMWBS, median split			<0.001
Below	130 (44%)	135 (64%)	
Above	165 (56%)	76 (36%)	
Missing	1	2	
Self-presentation, average summed score	1.6 (0.7)	2.2 (0.8)	<0.001
Missing	15	9	

^1^ Statistics presented: n (%); mean (SD). ^2^ Statistical tests performed: chi-square test of independence; Fisher’s exact test; Wilcoxon rank-sum test. ^a^ Response categories “a lot” and “very much” collapsed to avoid cells <5.

## Data Availability

The Norwegian Health research legislation and the Norwegian Ethics committees require explicit consent from the participants in order to transfer health research data outside of Norway. For the data employed in the present study, ethics approval was also contingent on storing the research data on secure storage facilities located at the Norwegian Institute of Public Health, which prevents us from providing the data as supplementary information or to transfer it to data repositories. Individual requests for data access should be sent to research project leader: jens.christoffer.skogen@fhi.no.

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
