# Peer review of "Through the Looking Glass of Social Media. Focus on Self-Presentation and Association with Mental Health and Quality of Life. A Cross-Sectional Survey-Based Study"

_ijerph, 2021, doi:10.3390/ijerph18063319_

Round 1

Reviewer 1 Report

First of all, I would like to congratulate the researchers for their work. The objective is very appropriate: "to investigate the association between focus on self-presentation on social media and mental health and quality of life among adolescents". This objective is in line with the social and technological changes of the society in which we live. In addition, it highlights the growing use and greater "dependence" on social networks, possible benefits and problems derived from them. All this, specifically in a group that due to their age could be more influenced.

I consider that this objective is well executed, from the introduction, which also acts as a brief review of the literature, through the methodology, to the discussion and conclusions. Moreover, the article is well structured and well written. Furthermore, I believe that the conclusions derived from the article, without being "groundbreaking", represent a contribution that can benefit future research, especially in view of the novelty of the subject matter addressed. For all these reasons, the article should be accepted.

However, like all research, it has limitations, some pointed out by the authors, others not so much. From here on, I will try to expose those aspects that I think could contribute to improve the final result:   

a) Introduction

Lines 43 and 44: “as some studies indicate that social media use is associated with more mental health problems and decreased well-being”

What studies, what problems, and briefly what their conclusions were. Although the authors, starting with line 73, make an overview of the research that deals with "self-presentation on social media", it would be interesting to provide the reader with background information on other variables.

For example: from classic contributions such as “Kraut, R., Patterson, M., Lundmark, V., Kiesler, S., Mukophadhyay, T., & Scherlis, W. (1998). Internet paradox: A social technology that reduces social involvement and psychological well-being?. American psychologist53(9), 1017.”

Until much more recent studies

-Hunt, M., All, K., Burns, B., & Li, K. (2021). Too Much of a Good Thing: Who We Follow, What We Do, And How Much Time We Spend on Social Media Affects Well-Being. Journal of Social and Clinical Psychology40(1), 46-68.

-Jarman, H. K., Marques, M. D., McLean, S. A., Slater, A., & Paxton, S. J. (2021). Motivations for Social Media Use: Associations with Social Media Engagement and Body Satisfaction and Well-Being among Adolescents. Journal of Youth and Adolescence, 1-15.

b) Materials and methods

Between lines 353-356 the authors state "Second, the questions related to focus on self-presentation on social media were specifically developed for the purposes of the survey. Specifically, the questions related to focus on self-presentation were derived from four focus-groups where social media, mental health and well-being was the over arching topic.", as a strength of the work. I think they should include in section 2.2.1 (line 152) more information on how these questions were arrived at, for example, who made up these groups, and why they did not use some already validated scale.

For example, they could indicate that they focused on some specific questions related to the "The Self-Presentation on Facebook Questionnaire (SPFBQ; Michikyan, Subrahmanyam, & Dennis, 2014) is a 17-item scale that measures varying degrees of presentation of the self on Facebook. The questionnaire assesses the degree in which an individual expresses facets of the real, ideal, and false self on Facebook with a 5-point Likert scale (1 strongly disagree, to 5 strongly agree)." (Jackson & Luchner, 2018)

-Jackson, C. A., & Luchner, A. F. (2018). Self-presentation mediates the relationship between self-criticism and emotional response to Instagram feedback. Personality and Individual Differences, 133, 1-6. 

On the other hand, the authors should justify the statistical analyses employed in line with the objectives. This could be done in section 2.3, specifically pointing out any study that applies this type of analysis in the field of social networks or similar.

c) Results

In this section, the authors could expand the comments on Figure 1, and especially on Figure 2, this would help the reader to better understand the results.

d) Discussion

The authors are aware of some important limitations of the work. In my point of view, we do not know important variables that could be determinant in the results achieved (this is the main weakness of this work). Above all, the profile of the respondents as users of social networks, number of networks they use, which ones (Facebook, Instagram...)? And specifically, as the authors themselves acknowledge, how much time do they spend on social networks? This aspect is pointed out both in the introduction "However, most of the studies investigating this association focus primarily on frequency and duration of use [5-7]", and in the limitations "Third, the survey did not include measures of frequency or duration of social media use among adolescents. However, such measure are likely to be inaccurate and biased [50, 51], and may be non-germane [5-7, 20]", opens the door to future lines of research that may refine this first descriptive analysis.

In this sense, I suggest that the authors include a specific subsection on future lines of research in which they present their proposals for solving this problem, and briefly point out some relationships that appear in the literature with respect to the time spent by the user on social networks.

Finally, I am looking forward to reading the final version of this paper, which I hope will be accepted once the proposed changes have been made.

Author Response

First of all, I would like to congratulate the researchers for their work. The objective is very appropriate: "to investigate the association between focus on self-presentation on social media and mental health and quality of life among adolescents". This objective is in line with the social and technological changes of the society in which we live. In addition, it highlights the growing use and greater "dependence" on social networks, possible benefits and problems derived from them. All this, specifically in a group that due to their age could be more influenced.

I consider that this objective is well executed, from the introduction, which also acts as a brief review of the literature, through the methodology, to the discussion and conclusions. Moreover, the article is well structured and well written. Furthermore, I believe that the conclusions derived from the article, without being "groundbreaking", represent a contribution that can benefit future research, especially in view of the novelty of the subject matter addressed. For all these reasons, the article should be accepted.

Response #1: We thank the reviewer for the summary and positive comments.

However, like all research, it has limitations, some pointed out by the authors, others not so much. From here on, I will try to expose those aspects that I think could contribute to improve the final result:   

  1. a) Introduction

Lines 43 and 44: “as some studies indicate that social media use is associated with more mental health problems and decreased well-being”

What studies, what problems, and briefly what their conclusions were. Although the authors, starting with line 73, make an overview of the research that deals with "self-presentation on social media", it would be interesting to provide the reader with background information on other variables.

For example: from classic contributions such as “Kraut, R., Patterson, M., Lundmark, V., Kiesler, S., Mukophadhyay, T., & Scherlis, W. (1998). Internet paradox: A social technology that reduces social involvement and psychological well-being?. American psychologist53(9), 1017.”

Until much more recent studies

-Hunt, M., All, K., Burns, B., & Li, K. (2021). Too Much of a Good Thing: Who We Follow, What We Do, And How Much Time We Spend on Social Media Affects Well-Being. Journal of Social and Clinical Psychology40(1), 46-68.

-Jarman, H. K., Marques, M. D., McLean, S. A., Slater, A., & Paxton, S. J. (2021). Motivations for Social Media Use: Associations with Social Media Engagement and Body Satisfaction and Well-Being among Adolescents. Journal of Youth and Adolescence, 1-15.

Response #2: We agree that some broader background information may be beneficial, and have now expanded this a bit in the revised background. We have also added the studies by Jarman et al. and Hunt et al., as well as some additional references throughout the text. We did, however, not include Kraut et al. as it did not seem to be directly relevant for our particular focus.

  1. b) Materials and methods

Between lines 353-356 the authors state "Second, the questions related to focus on self-presentation on social media were specifically developed for the purposes of the survey. Specifically, the questions related to focus on self-presentation were derived from four focus-groups where social media, mental health and well-being was the over arching topic.", as a strength of the work. I think they should include in section 2.2.1 (line 152) more information on how these questions were arrived at, for example, who made up these groups, and why they did not use some already validated scale.

For example, they could indicate that they focused on some specific questions related to the "The Self-Presentation on Facebook Questionnaire (SPFBQ; Michikyan, Subrahmanyam, & Dennis, 2014) is a 17-item scale that measures varying degrees of presentation of the self on Facebook. The questionnaire assesses the degree in which an individual expresses facets of the real, ideal, and false self on Facebook with a 5-point Likert scale (1 strongly disagree, to 5 strongly agree)." (Jackson & Luchner, 2018)

-Jackson, C. A., & Luchner, A. F. (2018). Self-presentation mediates the relationship between self-criticism and emotional response to Instagram feedback. Personality and Individual Differences, 133, 1-6.

Response #3: We thank the reviewer for pointing this out, and suggesting potential references. A paper giving a more detailed description of the main themes and results from the focus group interviews are currently under review in another journal and the pre-print is available here: https://www.researchsquare.com/article/rs-133641/v1.

We have also expanded on this in the revised manuscript in an effort to provide a short self-contained description of the process from the focus group interviews to the final questions that ended up in the survey. In the revised manuscript the start of section 2.2.1. reads as follows:

The majority of the questions regarding focus on self-presentation on social media were generated using an inductive approach [36], while one question came from an existing survey [37]. The remaining items developed were based on four focus group interviews with adolescents related to use of social media and mental health. The transcripts of the interviews were analysed using thematic analysis, and items were generated to cover the identified themes (more details can be found in the online preprint: https://www.researchsquare.com/article/rs-133641/v1). To create items that sounded familiar to the respondents (of the questionnaire), items were worded as similar to what the interview subjects expressed as possible. The use of focus groups ensure that the instrument is relevant to the population in question and that issues considered important for adolescents were not left out, i.e. they ensure the content validity of the instrument [38]. Candidate items were reviewed and discussed by members of the research team before being included in the questionnaire. Next, a small group of psychology students and a resource group of consisting of adolescents completed a draft of the questionnaire and provided oral feedback to the proposed questions. The questionnaire was revised according to their feedback, which mainly pertained to the wording of some of the items. One of the emergent sub-themes identified was self-presentation on social media. Participant’s focus on self-presentation on social media was measured by seven questions:

  1. I use a lot of time and energy on content I post on social media
  2. It is important for me that my posts receive many likes and/or comments
  3. It is important for me to have many followers on social media
  4. I delete posts on social media that do not receive enough likes and/or comments
  5. I retouch pictures of myself to look better before I post them on social media
  6. What other’s post (images/status updates/stories) makes me less satisfied with myself and my life
  7. The response I get for what I post (images/status updates/stories) impacts how I feel

The response categories were “not at all”, “very little”, “sometimes/partly true”, “a lot” and “very much”, coded 1 to 5. The questions are used separately and as a summed mean score (Cronbach’s alpha of 0.88) in the present study.

This comment is also related to a comment made by reviewer #2. Collectively, these comments led us to take one step back and have a more thorough look at items included in the present study which specifically could relate to self-presentation on social media. Upon re-assessing the available items, we decided to include the following two additional items which in their content relate to focus on self-presentation:

  • What other posts (images/status updates/stories) makes me less satisfied with myself and how I feel
  • The response I get for what I post (images/status updates/stories) impacts how I feel

After inclusion of these additional items we performed an exploratory factor analysis (EFA), as well as a confirmatory factor analysis (CFA) to determine the structural validity of the items in relation to dimensionality. Based on the EFA we found strong indications that all of the items could be thought of as belonging to unidimensional construct. The CFA confirmed this, and yielded good model fit indices for a 1-factor solution containing all seven items. Thus it was decided to retain all seven items in the revised manuscript. The inclusion of the two additional items did not change our main conclusions or interpretations, but strengthened our findings and provided more robust estimates. The EFA and CFA is described in more detail under “Statistical analyses” and “Results” in the revised manuscript.

On the other hand, the authors should justify the statistical analyses employed in line with the objectives. This could be done in section 2.3, specifically pointing out any study that applies this type of analysis in the field of social networks or similar.

Response #4: We are not sure what justification the reviewer is referring to here. Mean comparisons, standardized mean difference estimations and linear regression models are frequently used methods within many fields where quantitative methods are used, including social sciences and public health research. We do, however, acknowledge that more information regarding the use of blobbograms may be serviceable, as they are more frequently used in meta-analysis than in primary research. We have therefore tried to justify the statistical analyses more thoroughly in the revised manuscript. This section now reads:

Descriptive gender-specific statistics of the independent and dependent variables are presented in Table 1.  In Figure 1, the associations between each self-presentation item and symptoms of anxiety and depression and quality of life are presented as blobbograms [42]. Blobbograms were used as they provide information about individually estimated associations, the precision of the estimates, and statistical significance, as well as an over-all effect estimate in a succinct manner. This approach is widely used in meta-analysis but also in health research studies in general [42]. For Figure 1 the mean of each self-presentation item for those below and above the cut-offs for anxiety, depression and quality of life was calculated, as well as the standardised mean difference.

In order to assess the structural validity of the items related to focus on self-presentation, exploratory (EFA) and confirmatory factor (CFA) analysis was used to examine the underlying structure. Maximum likelihood (ML) estimation was used for CFA, and model fit was assessed by the comparative fit index (CFI), the root mean square error of approximation (RMSEA), and the standardized root mean square residual (SRMR). A CFI of 0.95 or greater, RMSEA of 0.06 or less, and SRMR of 0.08 or less were collectively considered an indication of good model fit to data [43].

The overall standardised mean difference for self-representation as a summed mean score is presented for each dependent variable. In Figure 2, the predicted mean gender-standardised score for  symptoms of depression and anxiety and quality of life (QoL) is presented across integers of the summed mean score of self-presentation for boys and girls separately. Linear gender-specific regressions were calculated to obtain the coefficient for the association between the mean summed score of self-presentation and the dependent variables. The mean gender-standardized score for symptoms of anxiety and depression, and QoL was obtained by a gender-stratified Z-scoring of the summed variables (total score). Thus, each gender-specific bar represents deviance in standard deviations from overall gender-specific mean across focus on self-presentation, and the error bars denotes 95% confidence intervals. The variables included in the present study had some missing values, ranging from n=3 to n=30. To retain the maximum level of information in the analyses employed in Figure 1 and 2, we employed pairwise deletion to handle missingness.The statistical analyses were performed using Stata 15 [44], R [45] and RStudio [46] including the R-packages “gtsummary” [47], “lavaan” [48], and “meta” [49].

  1. c) Results

In this section, the authors could expand the comments on Figure 1, and especially on Figure 2, this would help the reader to better understand the results.

Response #5: We thank the reviewer for pointing out that more details would be serviceable. We have now added some additional information on figure 1 and figure 2.

  1. d) Discussion

The authors are aware of some important limitations of the work. In my point of view, we do not know important variables that could be determinant in the results achieved (this is the main weakness of this work). Above all, the profile of the respondents as users of social networks, number of networks they use, which ones (Facebook, Instagram...)? And specifically, as the authors themselves acknowledge, how much time do they spend on social networks? This aspect is pointed out both in the introduction "However, most of the studies investigating this association focus primarily on frequency and duration of use [5-7]", and in the limitations "Third, the survey did not include measures of frequency or duration of social media use among adolescents. However, such measure are likely to be inaccurate and biased [50, 51], and may be non-germane [5-7, 20]", opens the door to future lines of research that may refine this first descriptive analysis.

In this sense, I suggest that the authors include a specific subsection on future lines of research in which they present their proposals for solving this problem, and briefly point out some relationships that appear in the literature with respect to the time spent by the user on social networks.

Response #6: We thank the reviewer for these comments and we hope that our suggested revisions are perceived as improvements in that respect. In relation to type of platform the participants used, we have some information about this which we now have included in the revised manuscript. We asked the participants which platforms they use from a list of 19 available options, and we have now included the ten most frequently reported used platforms in the revised manuscript. Specifically, we included this information in an analyses of differential use of platforms across quartiles of the summed self-presentation variable. Please refer to the revised manuscript for this change (changes made to the methods, results and discussion part of the paper).

We have also included a new section highlighting potential avenues for future research which may be important to explore (under 4.4. Future research and scope):

Given the preliminary nature of the present findings, our findings needs to be investigated in other samples. Specifically, there is a need to investigate the proposed associations in larger and more diverse study samples. This would increase the precision of the estimates as well as enable investigations into potential confounding and interaction effects. Longitudinal data would also be beneficial, as this could get us a step closer to disentangling the temporal association between self-presentation and mental health and well-being, as well as open up for an exploration of mediation effects. The present study demonstrate the advantage of considering specific aspects of social media use, unravelling the diverse effects of social media use as indicated by conflicting previously reported study results [30].

Finally, I am looking forward to reading the final version of this paper, which I hope will be accepted once the proposed changes have been made.

Reviewer 2 Report

The manuscript presents a correlational study between social network use and self-presentation and mental health and well-being of adolescents in Norwegian schools. It is a correctly and comprehensibly structured study, although the following changes are suggested:

  1. Read and use the manuscript: Núñez, A., Álvarez-García, D., & Pérez-Fuentes, M. (2021). Anxiety and self-esteem in cyber-victimization profiles of adolescents. Comunicar, 67, 47-59. https://doi.org/10.3916/C67-2021-04
  2. Include a section on permissions and ethics in the section on materials and methods. In some countries (as is the case in Spain), permission from the parents or guardians of minors is required to carry out studies with this type of samples. It should be reported whether these permissions were obtained. 
  3. The authors should explain how the survey instrument was constructed, as well as its content and construct validity.

Author Response

The manuscript presents a correlational study between social network use and self-presentation and mental health and well-being of adolescents in Norwegian schools. It is a correctly and comprehensibly structured study, although the following changes are suggested:

  1. Read and use the manuscript: Núñez, A., Álvarez-García, D., & Pérez-Fuentes, M. (2021). Anxiety and self-esteem in cyber-victimization profiles of adolescents. Comunicar, 67, 47-59. https://doi.org/10.3916/C67-2021-04

Response #1: We thank the reviewer for this suggestion, however, we did not include it in the manuscript, as we felt it was somewhat beyond the scope of the present study, with its particular focus on cyber-victimization.

  1. Include a section on permissions and ethics in the section on materials and methods. In some countries (as is the case in Spain), permission from the parents or guardians of minors is required to carry out studies with this type of samples. It should be reported whether these permissions were obtained. 

Response #2: Thank you for pointing this out. The ethics statement is presently included under the section “Institutional Review Board Statement” as per the journal formatting instructions. We have, however, added a short sentence about ethics under materials and methods. We have also expanded the Institutional Review Board Statement to specifically address your concerns regarding consent competency in relation to age.

  1. The authors should explain how the survey instrument was constructed, as well as its content and construct validity.

Response #3: We thank the reviewer for this comment, which is also akin to a comment made by reviewer #1. These comments led us to take one step back and have a more thorough look at items included in the present study which specifically could relate to self-presentation on social media. Upon re-assessing the available items, we decided to include two additional items which in their content relate to focus on self-presentation:

  • What other’s post (images/status updates/stories) makes me less satisfied with myself and my life
  • The response I get for what I post (images/status updates/stories) impacts how I feel

After inclusion of these additional items we performed an exploratory factor analysis, as well as a confirmatory factor analysis to determine the structural validity of the items in relation to dimensionality. Based on the EFA we found strong indications that all of the items could be thought of as belonging to unidimensional construct. The CFA confirmed this, and yielded good model fit indices for a 1-factor solution containing all seven items. Thus it was decided to retain all seven items in the revised manuscript. The inclusion of the two additional items did not change our main conclusions or interpretations, but strengthened our findings and provided more robust estimates. The EFA and CFA is described in more detail under “Statistical analyses” and “Results” in the revised manuscript.

Although we have now more thoroughly assessed the structural validity of our items, it is more difficult to assess the criterion, construct and content validity of the items directly, as we do not have any benchmark for comparison in relation to “focus on self-presentation on social media”. We do, however, believe that the items included have a high face validity as they can readily be conceived of as being relevant for focus on self-presentation on social media. The lack of available comparative evidence in relation to the validity of the items used as a scale is one of the reasons we chose to also present the associations with our outcomes for the separate items in figure 1. Despite this, we now included a little more information about the process from which the questions arose – based on focus-group interviews (see below and methods in the revised manuscript).

We also agree that more details about the construction of the items is beneficial. The main themes and results from the focus group interviews are currently under review in another journal but a pre-print is available here: https://www.researchsquare.com/article/rs-133641/v1. We do, however acknowledge that more information regarding this should be available in the present study. We have therefore expanded on this in the revised manuscript in an effort to provide a short self-contained description of the focus group interviews. Now the start of section 2.2.1. reads as follows:

The majority of the questions regarding focus on self-presentation on social media were generated using an inductive approach [36], while one question came from an existing survey [37]. The remaining items developed were based on four focus group interviews with adolescents related to use of social media and mental health. The transcripts of the interviews were analysed using thematic analysis, and items were generated to cover the identified themes (more details can be found in the online preprint: https://www.researchsquare.com/article/rs-133641/v1). To create items that sounded familiar to the respondents (of the questionnaire), items were worded as similar to what the interview subjects expressed as possible. The use of focus groups ensure that the instrument is relevant to the population in question and that issues considered important for adolescents were not left out, i.e. they ensure the content validity of the instrument [38]. Candidate items were reviewed and discussed by members of the research team before being included in the questionnaire. Next, a small group of psychology students and a resource group of consisting of adolescents completed a draft of the questionnaire and provided oral feedback to the proposed questions. The questionnaire was revised according to their feedback, which mainly pertained to the wording of some of the items. One of the emergent sub-themes identified was self-presentation on social media. Participant’s focus on self-presentation on social media was measured by seven questions:

  1. I use a lot of time and energy on content I post on social media
  2. It is important for me that my posts receive many likes and/or comments
  3. It is important for me to have many followers on social media
  4. I delete posts on social media that do not receive enough likes and/or comments
  5. I retouch pictures of myself to look better before I post them on social media
  6. What other’s post (images/status updates/stories) makes me less satisfied with myself and my life
  7. The response I get for what I post (images/status updates/stories) impacts how I feel

The response categories were “not at all”, “very little”, “sometimes/partly true”, “a lot” and “very much”, coded 1 to 5. The questions are used separately and as a summed mean score (Cronbach’s alpha of 0.88) in the present study.
